# Evaporation-Rate Control of Water Droplets on Flexible Transparent Heater for Sensor Application

**DOI:** 10.3390/s19224918

**Published:** 2019-11-12

**Authors:** Jaesoung Park, Suhan Lee, Dong-Ik Kim, Young-You Kim, Samsoo Kim, Han-Jung Kim, Yoonkap Kim

**Affiliations:** 1Convergence Materials Research Center, Gumi Electronics & Information Technology Research Institute (GERI), Gumi 39171, Korea; jspark@geri.re.kr (J.P.); sskim@geri.re.kr (S.K.); 2Convergence Medical Devices Research Center, Gumi Electronics & Information Technology Research Institute (GERI), Gumi 39253, Korea; swan@geri.re.kr; 3Center for Integrated Smart Sensors (CISS), Korea Advanced Institute of Science and Technology (KAIST), Daejeon 34141, Korea; dikim915@kaist.ac.kr; 4Department of Physics, Kongju National University, Gongju 32588, Korea; yykim@kongju.ac.kr

**Keywords:** transparent heater, wettability, droplet evaporation, surface treatment

## Abstract

To develop high-performance de- or anti-frosting/icing devices based on transparent heaters, it is necessary to study the evaporation-rate control of droplets on heater surfaces. However, almost no research has been done on the evaporation-rate control of liquid droplets on transparent heaters. In this study, we investigate the evaporation characteristics of water droplets on transparent heater surfaces and determine that they depend upon the surface wettability, by modifying which, the complete evaporation time can be controlled. In addition, we study the defrosting and deicing performances through the surface wettability, by placing the flexible transparent heater on a webcam. The obtained results can be used as fundamental data for the transparent defrosting and deicing systems of closed-circuit television (CCTV) camera lenses, smart windows, vehicle backup cameras, aircraft windows, and sensor applications.

## 1. Introduction

Of late, high-performance transparent heaters have attracted considerable interest as de- or anti-frosting/icing windows and heating sources for automobiles, airplanes, displays, sensors, biochips, and greenhouses [1,2,3]. However, studies on transparent heaters have mainly focused on the advanced materials and manufacturing technologies required for realizing high performance, and their application in various fields [1,4,5,6,7,8,9,10]. For the commercialization of transparent heaters, fundamental research on heaters, including the physical and chemical characteristics of the heater surface and interface, must be established for high durability and performance.

Therefore, in this study, we fabricate a high-performance flexible transparent heater, which includes an indium tin oxide (ITO)/Ag/ITO multilayer on a polyethylene terephthalate (PET) substrate, using roll-to-roll (RTR) sputtering systems, and investigate the real-time evaporation characteristics of water droplets on the heater surface using surface treatment (hydrophilicity). Based on the results, we demonstrate that the water droplets on the hydrophilic surface of a transparent heater evaporate more rapidly with low power consumption, compared to the other surface states, due to the higher evaporation flux of the droplet, which is thinner and has a larger contact area with the heating surface of the heater. This indicates that the droplet evaporation time on the surface of the transparent heater can be controlled by the surface wettability, utilizing the temperature difference between the heater surface and ambient air. Thus, these results can be applied fundamental surface and interface analyses data for the manufacture of transparent de-icing/frosting systems for closed-circuit television (CCTV) camera lenses, smart windows, side-mirrors of automobiles, aircraft windows, and sensor applications.

## 2. Materials and Methods

### 2.1. ITO/Ag/ITO Structure Transparent Flexible Heater Fabrication 

We fabricated high-performance, flexible, transparent heaters with an ITO/Ag/ITO conductive multilayer fabricated using the continuous RTR sputtering method. Conductive multilayer films were continuously sputtered at room temperature on a PET substrate with a thickness of 125 um using an RTR sputtering system (RVS250, Max Film, Daegu, Korea). The bottom ITO layer with a thickness of 30 nm was sputtered onto the PET substrate using an ITO target. The sputtering conditions were as follows: Midrange frequency power of 1.5 kW, working pressure of 2.5 mTorr, Ar/O_2_ flow rate of 600/2.7 sccm, and rolling speed of 0.5 m/min. After sputtering the bottom ITO layer, an Ag inter-layer with a thickness of 13 nm was continuously sputtered onto the bottom ITO layer at a working pressure of 2.5 mTorr. After sputtering the Ag interlayer, a 30 nm thick top ITO layer was sputtered onto the Ag interlayer. The same sputtering conditions were used for the top and bottom ITO layers in order to form a symmetric multilayer structure.

### 2.2. Optical, Electrical, and Structural Characterization 

To investigate the optical properties, the transmittance spectra of the flexible transparent heaters were observed using a UV-VIS-NIR spectrophotometer (SolidSpec-3700, Shimadzu Scientific Instruments, city, country) with air as the reference. In order to confirm the structural properties and chemical composition of ITO/Ag/ITO multilayer transparent heater, transmission electron microscopy (TEM) (JEM-2100F HR, JEOL Ltd., Tokyo, Japan) and energy-dispersive X-ray spectroscopy (EDS) mapping analysis were performed. The cross-section TEM image of the fabricated transparent flexible heater was obtained using a field emission scanning electron microscope (FE-SEM) (Sirion, FEI/Philips, Hillsboro, OR, USA). The sheet resistance of the fabricated transparent heater was measured using the four-point probe method with a sheet resistivity meter (FPP-1000, DASOL ENG, Cheongju, Korea) in contact with the front-center of the heater. The sheet resistances, listed in this study, were typically obtained by averaging several measurements (at least five) performed at different positions on the front-side of the heater.

### 2.3. Evaluation of the Heat-Generation Performance of the Flexible Transparent Heater 

To assess the heat-generation performance of the fabricated flexible transparent heater, 20 mm × 20 mm ITO/Ag/ITO/PET films with a two-terminal side-contact configuration were fabricated. Direct current (DC) voltages of 0.5, 1.0, 1.5, 2.0, and 2.5 V, respectively, were applied to the fabricated heater through the Ag side-contact, and the resulting variation in temperature was measured by direct measurement with a thermocouple (ST-50, RKC instrument Inc., Tokyo, Japan) mounted on the back-side of the heater. The obtained temperature profile was confirmed by using an infrared (IR) camera (LT3-P, Zhejiang Dali Technology Co., Ltd, Hangzhou, China).

### 2.4. Oxygen (O_2_) Plasma Treatment for Wettability Control 

In order to modify the surface of the heater, oxygen (O_2_) plasma treatments were performed on the surface (PET side) of the heater using an O_2_ plasma generation system (LF PlasmaSTAR 100, JNE Corp., Suwon, Korea), under a working condition including 20 W power, 20 sccm oxygen flow, and 200 mTorr chamber pressure for 30 s.

### 2.5. Contact Angle Measurement 

The static contact angle (CA) of the droplet on the surface of the transparent heater was measured, before and after O_2_ plasma treatment, with a CA analyzer (Phoenix 300, SEO, Suwon, Korea), using a pipette to drop 2.0 ul deionized (DI) water. The cold light source used for backlighting ensured improved contrast without affecting the droplet evaporation time. The CAs, listed in this study, were typically obtained by averaging several measurements (at least three) performed at different parts of the heater surface. The CA measurements were performed at a room temperature of 23 °C and relative humidity of 45%.

### 2.6. Real-Time Monitoring System for Droplet-Evaporation Observation on the Heater

The droplet evaporation process on each flexible transparent heater surface was recorded by a digital microscope imaging system (Dino-Lite AD7013MZT, AnMo Electronics Corp., New Taipei, Taiwan) with a white light-emitting diode light source (Appendix A). To analyze the droplet evaporation, 4.0 μl of DI water was gently dropped onto the heater surface using a calibrated microsyringe. Further, a DC power supply (GPS-2303, Good Will Instrument Co., Ltd, New Taipei, Taiwan) was utilized for adjusting the heater temperature by varying the applied voltage. The temperature variation was monitored through direct measurement with a thermocouple mounted on the front-side of the heater. In addition, the entire droplet evaporation process was recorded at a resolution of 1280 × 960 pixels, and the time required for complete droplet evaporation was analyzed. The evaporation time was measured five times and averaged at different positions on the heater surface, under a temperature- and humidity-controlled environment.

## 3. Results and Discussion

In this study, ITO/Ag/ITO multilayer films fabricated by continuous RTR sputtering were used as flexible transparent heaters. Appendix A depicts the photograph and sheet resistance (4–6 Ω/sq) of the flexible transparent heater, which is fabricated on a PET substrate (100 × 100 mm^2^), with excellent optical and electrical properties. The presence of a 13 nm-thick Ag interlayer between the top (30 nm thick) and bottom (30 nm thick) ITO layers was confirmed by cross-sectional TEM and EDS images, as shown in Figure 1a. The optical transmittance spectra of the transparent heater (Figure 1b) exhibited an average transmittance of 80% over the entire visible range (400–800 nm) and transmittance of 86.5% at 550 nm. 

The figure of merit (FoM) value of the ITO/Ag/ITO multilayer transparent heater, which is the ratio of electrical to optical conductance ($\sigma_{dc}/\sigma_{opt}$), was calculated using the obtained experimental results, as follows [11,12]:(1)$$T\;=\;{\left(1+\frac{Z_{0}}{2R_{S}}\frac{\sigma_{\mathrm{opt}}}{\sigma_{\mathrm{dc}}}\right)}^{-2}$$where $R_{S}$ and T are the measured sheet resistance and transmittance at 550 nm, respectively, and $Z_{0}$ is the impedance of free space (377 Ω). The FoM value of 501.3, obtained by Equation (1), is 10 times greater than the minimum FoM value (50) required for electrical devices, such as touch screens and displays [13,14].

To assess the heating performance of the ITO/Ag/ITO multilayer transparent heater, a 20 mm × 20 mm heater was fabricated, and the temperature profiles were obtained by direct measurement at the back-side of the heater. As shown in Figure 2a, when DC voltages of 0.5, 1.0, 1.5, 2.0, and 2.5 V, respectively, were applied to the transparent heater through the two-terminal Ag side contact, the steady-state temperature increased from 33.5 °C to 127.1 °C with the applied voltage. From Appendix A, the steady-state temperature of the transparent heater increases with the increase in input voltage and decrease in heater size. This result indicates that the steady-state temperature depends upon the heater surface area (unit area) because the power per unit area dissipated in the layer decreased with an increase in the unit area. Hence, the heat power loss significantly increases with the increase in heater size at same applied voltage.

The temperature changes generated in the heater on applying a voltage of 2 V were investigated for observing the thermal reproducibility of the heater, for 30 days. A steady-state temperature of 96–98 °C was consistently maintained during the switching cycles, as seen in Figure 2b and Appendix A. In general, the performances of devices with pure copper (Cu) and Ag electrodes or heaters are degraded by thermal oxidation during operation [9,13,15,16]. However, the thermal reproducibility results of the ITO/Ag/ITO multilayer transparent heater demonstrated that the heat generation performance of the heater was not affected by the 13 nm thick Ag interlayer between the top and bottom ITO layers, by thermal oxidation due to the larger heat transfer coefficient of the oxide layers.

Figure 3 displays the photograph and infrared (IR) images of the ITO/Ag/ITO multilayer transparent heater attached to the built-in webcam of a laptop, during operation at 50 °C. For the uniformly heated surface of the heater, the IR images without any contacts underneath the heater, as shown in Appendix A. The images captured by the webcam are depicted without (Appendix A) and with the transparent heater (Appendix A), and during operation at 50 °C (Appendix A). There are no differences in these figures, indicating that the transparent heater does not have any effect on the images captured by the webcam, even when the heater was operational. 

In addition, deicing and defrosting tests were performed with the ITO/Ag/ITO multilayer transparent heater attached to the built-in webcam of a laptop. A lump of ice placed on the heater completely disappeared in 4 min, when a temperature of 100 °C was provided. Images could then be clearly captured, as seen in Figure 4. Appendix A depicts the defrost test result of the ITO/Ag/ITO multilayer transparent heater after frost formation. The heater was placed in a refrigerator for 15 min for uniform frost formation on its entire surface. At an operating voltage of 1 V and a temperature of 50 °C, the frost on the surface was removed within 20 s. Hence, the ability of the webcam to capture clear images was not affected. The remaining water droplets on the surface were completely evaporated within 78 s.

Figure 5 displays the CA and photographs of the water droplets on the surface (PET side) of the ITO/Ag/ITO multilayer transparent heater fabricated on a PET substrate, before and after surface treatment with O_2_ plasma. The average droplet CA was approximately 94° (hydrophobic state) on the surface of the heater. The CA on the surface decreased from the initial value to 11° (hydrophilic state) with O_2_ plasma treatment. The transition from the hydrophobic (CA = 94°) to hydrophilic state (CA = 11°) indicates that the surface wettability of the heater, and the adhesion between the droplet and surface are significantly improved [17,18,19]. The hydrophilic surface was sustained, even after the thermal reproducibility test for 30 days. Thus, the reliability and performances of various devices with transparent heaters having hydrophilic substrates can be enhanced by improving the adhesion between the substrate and the active layer.

Figure 6 shows the microscopic images during the complete evaporation of a 4.0 μL water droplet on the surface (at room temperature of 23 °C), with surface treatment (hydrophilic state) and without (hydrophobic state). In the initial state, the water droplet on the hydrophobic surface of the transparent heater was nonwetting on the surface, projecting background images distorted by the refraction of light through the droplet. On the other hand, there was almost no distortion of the projected image on the hydrophilic surface due to the spreading of the water droplet, with surface treatment. As shown in Figure 6, the droplet on a hydrophilic surface has a shorter evaporation period than that on a hydrophobic surface. There are two main causes for this: One is the pinning phenomenon, wherein the contact radius of the droplet remains fixed during droplet evaporation (constant contact radius (CCR)). Generally, the pinning time reduces with the increase in the total evaporation time, as the surface hydrophobicity becomes strong [20]. In the case of the hydrophilic surface (with surface treatment) in Figure 6, the pinning phenomenon occupied approximately 88% of the total evaporating time, whereas the pinning phenomenon on the hydrophobic surface occupied approximately 40%, as shown in Figure 7. The other reason is that the evaporation flux along the surface of the droplet affects the total evaporation time. The average evaporation flux can be calculated using Equation (2) [21,22]:(2)$$J_{\mathrm{avg}}=\left(\rho \frac{dV}{dt}\right)/A$$where ρ, A, V and dV/dt are the density, surface area, volume, and rate of mass loss of the droplet, respectively. The surface area of the droplet on a hydrophobic surface is greater than that on a hydrophilic surface. Hence, a droplet evaporating on a hydrophilic surface has higher evaporation flux compared to that on a hydrophobic surface. Therefore, the total evaporation time of a droplet on a hydrophilic surface is lesser than that on a hydrophobic surface.

Figure 7 shows the microscopic images during the complete evaporation of a 4.0 μL water droplet on heated substrates (temperature of 95 °C), with and without surface treatment. The required time for complete evaporation on heated substrates was lesser than that required for non-heated substrates because of the higher evaporation flux on the droplet surface (this phenomenon also occurred at various heating temperatures (30, 45, 65 °C), as shown in Appendix A).

In addition, as shown in Figure 8, the difference in the complete evaporation time between hydrophobic and hydrophilic heater surfaces was considerably reduced with the increase in heating temperature. This can be explained by the evaporation flux distribution on the droplet. When the edge evaporation flux is greater than that on the other areas of the droplet, the complete evaporation time of the droplet is generally reduced because most of the liquid atoms break away from the droplet by surface diffusion, and evaporate to vapor atoms near the edge of the droplet [23]. 

Moreover, the edge of the droplet has a shorter distance to the substrate than the other areas of the droplet. As a result, the edge is hotter and has higher evaporation flux than the other surfaces of the droplet on the heating substrate. In the case of droplet evaporation on a hydrophobic substrate, the evaporation fluxes at the droplet edge are greater than those elsewhere on the droplet surface with the increase in heating temperature, in contrast to the lower evaporation fluxes at the edge of a non-heated substrate [24]. Therefore, the difference in the time required for the complete evaporation of the water droplet, between hydrophobic and the hydrophilic heater surfaces, was considerably reduced with the increase in temperature of the heater surface. Furthermore, it was observed that the total evaporation flux of the water droplet on a hydrophilic heater surface was greater than that on a hydrophobic surface because on the hydrophilic surface, the droplet is thinner and larger contact area with the heating surface of the heater. This phenomenon can be also explained based on the height (h)-to-contact radius ($R_{C}$) aspect ratio ($h/R_{C}$) of the surface droplet [21]. As shown in Figure 5, the aspect ratios of droplets on hydrophobic and hydrophilic surfaces are 0.56 and 0.04, respectively. For the hydrophilic surface, the droplet on a heated surface has shorter conduction path (h) and larger conduction base area ($A_{C}={\pi R_{C}}^{2}$) compared to the hydrophobic surface. Hence, the temperature drop across the droplet on a hydrophilic surface is considerably lower than that on a hydrophobic surface, reducing the complete evaporation time. In addition, the complete evaporation time with a lower ($h/R_{C}$) aspect ratio (lower CA) droplet on a heated surface can be reduced by increasing the difference between the temperature of the heated substrate and the ambient temperature, as shown in Equation (3) [21,24,25].
(3)$$T_{\mathrm{tot}}=\;\frac{\rho_{L}}{2D\left[c_{S}\left(T_{S}\right)-Hc_{S}\left(T_{a}\right)\right]}{\left(\frac{3V_{i}}{\pi }\right)}^{2/3}\frac{1}{{\left[g\left(\theta \right)\right]}^{1/3}f\left(\theta \right)}$$where $T_{\mathrm{tot}}$ is the time taken for complete evaporation, $\rho_{L}$ is the liquid density, D is the coefficient of vapor diffusion, $c_{S}$ is the saturated vapor concentration on the droplet surface, $T_{S}$ is the substrate temperature, $T_{a}$ is the ambient temperature, H is the far-field relative humidity, and θ is the CA of the liquid. Therefore, these results demonstrate that the optical characteristics and complete evaporation time of the transparent heater with droplets can be controlled by the surface wettability, and temperature difference between the heater surface and ambient air.

## 4. Conclusions

In this study, we investigated the heating performance of an ITO/Ag/ITO flexible transparent heater with hydrophilic treatment, for deicing and defrosting systems. The experimental results demonstrated that due to the low aspect ratio (lower CA) of the droplet on the hydrophilic surface of the ITO/Ag/ITO multilayer transparent heater, the water droplet rapidly evaporated with low power consumption, and there was almost no distortion and obstruction of the field-of-view by the droplets formed on the transparent heater surface. These results establish that the ITO/Ag/ITO multilayer flexible transparent heater with a hydrophilic surface can be used as a transparent de-icing/frosting device and heating source for the windows and side-mirrors of automobiles, camera lenses of CCTVs and drones, smart windows, and greenhouses. In addition, if the ITO/Ag/ITO multilayer transparent heater (or electrode) with a hydrophilic surface is applied to sensors, biochips, or solar cells, improvements in the durability and performance are expected due to the high adhesion and chemical affinity between the heater (or electrode) and the activity layer.

## Figures and Tables

**Figure 1 sensors-19-04918-f001:**
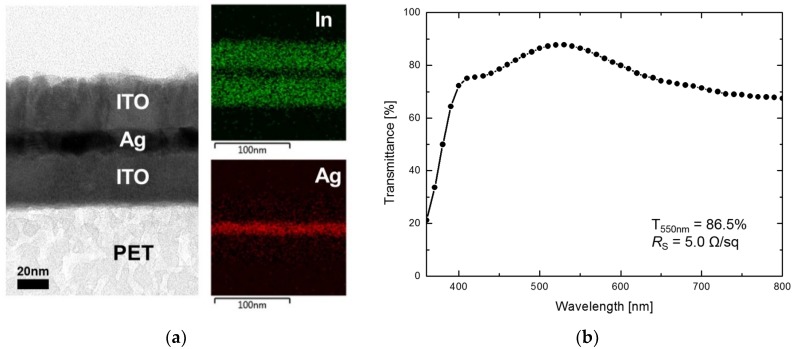
(**a**) Cross-sectional transmission electron microscopy (TEM) and energy-dispersive X-ray spectroscopy (EDS) images of the ITO/Ag/ITO multilayer transparent heater, and its (**b**) optical transmittance spectrum and sheet resistance.

**Figure 2 sensors-19-04918-f002:**
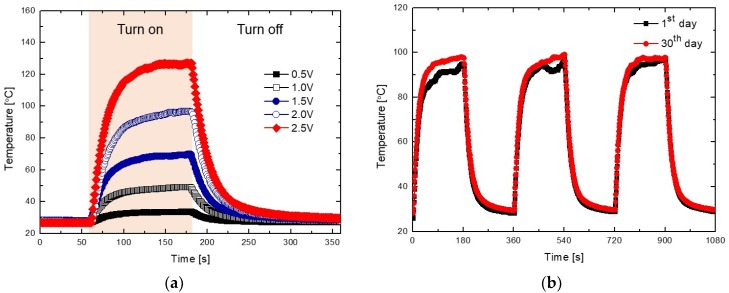
(**a**) Heat-generation performance of the ITO/Ag/ITO multilayer transparent heater as a function of time at applied voltages of 0.5, 1.0, 1.5, 2.0, and 2.5 V, respectively, and its (**b**) heat-generation reproducibility for 30 days.

**Figure 3 sensors-19-04918-f003:**
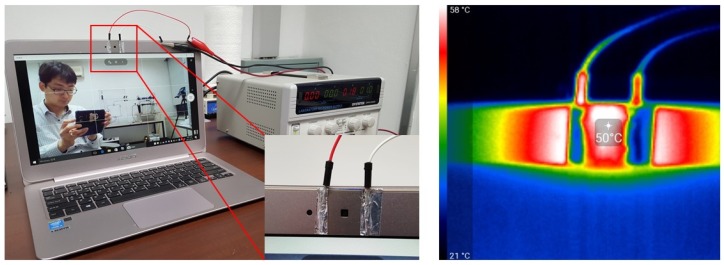
Photograph and infrared (IR) images of the ITO/Ag/ITO multilayer transparent heater attached to the built-in webcam of a laptop.

**Figure 4 sensors-19-04918-f004:**
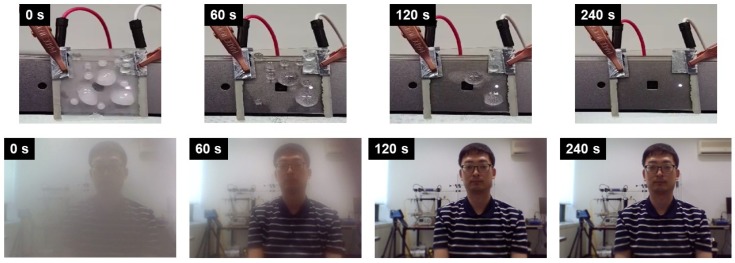
Photograph images of the deicing test results.

**Figure 5 sensors-19-04918-f005:**
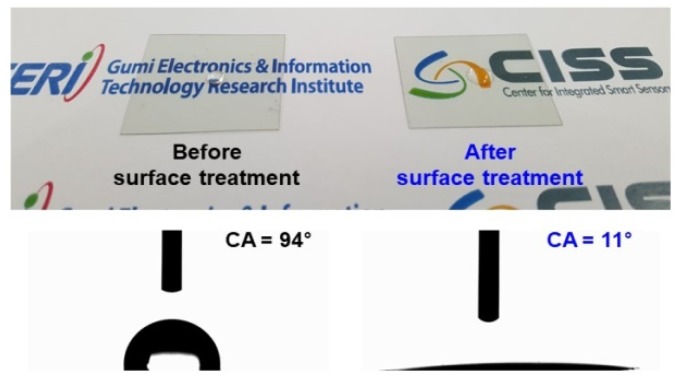
Photograph and contact angle (CA) images of deionized (DI) water droplets on the surface (PET-side) of the flexible transparent heater before and after surface treatment with O_2_ plasma.

**Figure 6 sensors-19-04918-f006:**
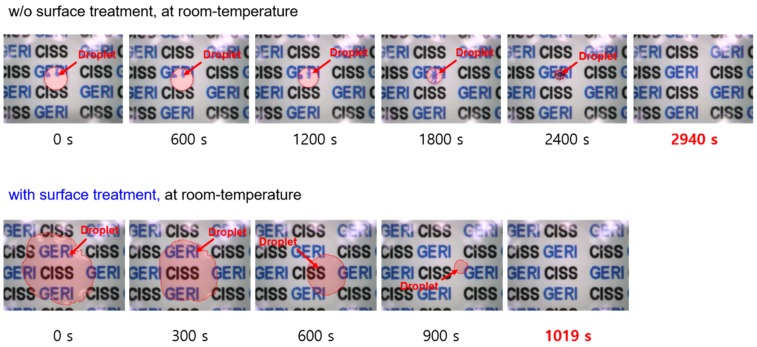
Microscopic images of DI water droplet evaporation on heater surfaces (without heating).

**Figure 7 sensors-19-04918-f007:**
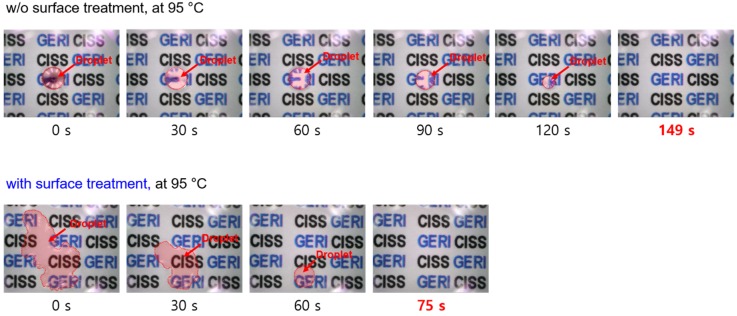
Microscopic images of DI water droplet evaporation on heater surfaces (heat-generation temperature of 95 °C).

**Figure 8 sensors-19-04918-f008:**
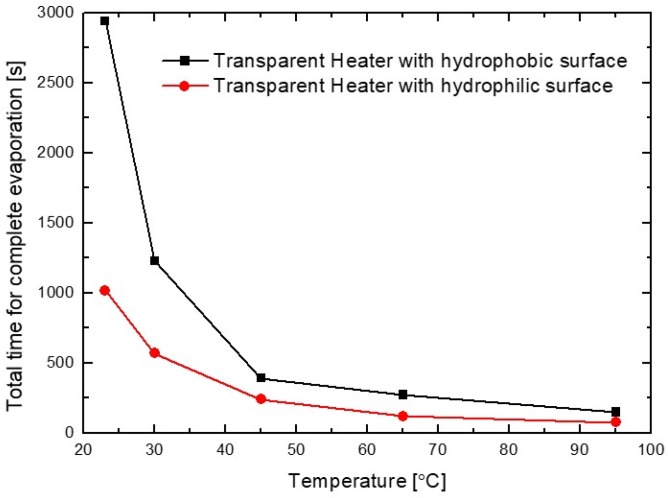
Total duration for complete water droplet evaporation on ITO/Ag/ITO multilayer transparent heater surfaces with different wetting properties and heating temperatures (room-temperature, 30, 45, 65, and 95 °C).

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
