# Peer review of "Evaporation-Rate Control of Water Droplets on Flexible Transparent Heater for Sensor Application"

_sensors, 2019, doi:10.3390/s19224918_

Round 1

Reviewer 1 Report

The authors tried to control the evaporation-rate of liquid droplets on transparent heaters by modifying the surface of substrate of conducting transparent layer.

In Abstract they have stated:

The obtained results can be used as fundamental data for the transparent defrosting and deicing systems of closed-circuit television (CCTV) camera lenses, smart windows, vehicle backup cameras, aircraft windows, and sensor applications.”

Unfortunately it is difficult to agree with this claim because of many numerous errors, simplifications and ambiguities in the manuscript presented.

The main claim are listed below.

Q1) Simple geometry (square) of the heating structure under study, therefore:

The heating of this structure is non-uniform, what is shown also by the authors (Figure 3.). The temperature measurement have been performed by one thermocouple, probably placed in the center of structure, what is unreliable.

Q2) The authors did not take into account the changing of electrical resistance of ITO/Ag/ITO layer due to heating.

Q3) Line 133-137 of the text: ” This result indicates that the steady-state temperature depends upon the heater surface area; hence, the heat power loss significantly increases with the increase in heater size at same applied voltage.”

Decreasing of steady-state temperature with increasing of heater size is caused by different value of surface power. The samples (10x10mm), (20x20mm), (30x30mm) have the same resistance but the power per unit area dissipated in the layer (by the same voltage) is not equal and this is main reason of differences between their effective heating.

Q4) Discussion about evaporation rate on the basis of images relating to the surface without modification (Figure 6, Figure 7, Figure S7-S9) is pointless. The pictures are unreadable!

Q5) The driving force for evaporation is heating. It is clear that the droplet on the PET-surface without treatment with the plasma is heated not so effective due to much smaller contact surface with the heating layer than in the case of the processed surface (see the Figure 5.). Only after considering this fact would the discussion about “pinning time” and “the evaporation flux” make sense (line 193-207).

Q6) Many of statements were to be expected or are obvious.

Author Response

Reviewer 1

Q1) Simple geometry (square) of the heating structure under study, therefore:

The heating of this structure is non-uniform, what is shown also by the authors (Figure 3.). The temperature measurement have been performed by one thermocouple, probably placed in the center of structure, what is unreliable.

 (Answer) We appreciate the valuable recommendation and comments of the referee. As the referee pointed out, we placed one thermocouple on the center of the sample for temperature measurement. Thus, we tried to display the infrared (IR) images of the transparent heater to show the temperature distribution. However, the IR image of the heating surface is non-uniform due to the heat loss from the surface of the laptop. Therefore, we measured the temperature of the heater with the IR images without any contacts underneath the heater. The results showed the uniformly heated area. We added revised sentence to line 152-154, and the IR images to supplementary material (Figure S5).

[Revised Sentence]

Line 135-137:

For the uniformly heated surface of the heater, the IR images without any contacts underneath the heater as shown in Figure S5.

[Added Figure S5]

Figure. S5. IR Images without any contacts underneath the heater.

Q2) The authors did not take into account the changing of electrical resistance of ITO/Ag/ITO layer due to heating.

 (Answer) As the referee’s comment, we agree that the changing of electrical resistance of ITO/Ag/ITO heater due to heating should be considered. Thus, we checked the changes of the electrical resistance before and after applying voltage of 6V for 60 minutes. The result showed there was no change in the electrical resistance of the heater due to heating.

< Electrical resistance of the ITO/Ag/ITO heater before (L) and after (R) heating >

Q3) Line 133-137 of the text: ” This result indicates that the steady-state temperature depends upon the heater surface area; hence, the heat power loss significantly increases with the increase in heater size at same applied voltage.”

Decreasing of steady-state temperature with increasing of heater size is caused by different value of surface power. The samples (10x10mm), (20x20mm), (30x30mm) have the same resistance but the power per unit area dissipated in the layer (by the same voltage) is not equal and this is main reason of differences between their effective heating.

 (Answer) The referee mentioned that the samples have the same resistance. If it means that the samples have the same sheet resistance, we agree with the referee’s comment. Actually, we tried to described the heat power loss increases with the increased in heater area (size) which meant the power per unit area dissipated in the layer decreased with an increase in the unit area. Thus, we revised the sentence in line 135-137 of the text.

[Revised Sentence]

Line 135-137:

This result indicates that the steady-state temperature depends upon the heater surface area(unit area) because the power per unit area dissipated in the layer decreased with an increase in the unit area; hence, the heat power loss significantly increases with the increase in heater size at same applied voltage

Q4) Discussion about evaporation rate on the basis of images relating to the surface without modification (Figure 6, Figure 7, Figure S7-S9) is pointless. The pictures are unreadable!

(Answer) As the referee’s comment, we modified the images relating to the surface to demonstrate the complete evaporation time (evaporation rate). Therefore, we added the revised the Figure 6, 7, Figure S7-S9 and sentences as below:

[Revised Figures]

Figure 6

Figure 7

[Revised Sentence]

Line 22:

the complete evaporation time can be controlled

Line 44-46:

This indicates that the droplet evaporation time on the surface of the transparent heater can be controlled by the surface wettability, utilizing the temperature difference between the heater surface and ambient air

Line 44-46:

The cold light source used for backlighting ensured improved contrast without affecting the droplet evaporation time.

Line 247-248:

Hence, the temperature drop across the droplet on a hydrophilic surface is considerably lower than that on a hydrophobic surface, reducing the complete evaporation time.

Line 255-257:

Therefore, these results demonstrate that the optical characteristics and complete evaporation time of the transparent heater with droplets can be controlled by the surface wettability, and temperature difference between the heater surface and ambient air.

Q5) The driving force for evaporation is heating. It is clear that the droplet on the PET-surface without treatment with the plasma is heated not so effective due to much smaller contact surface with the heating layer than in the case of the processed surface (see the Figure 5.). Only after considering this fact would the discussion about “pinning time” and “the evaporation flux” make sense (line 193-207).

 (Answer) In the Equation (2) for the evaporation flux, the surface area (A) is not contact surface with the heating layer but the surface area of the droplet. Therefore, the surface area of the droplet on the hydrophobic surface (without plasma treatment) is greater than that on a hydrophilic surface. Finally, the complete evaporation time of the droplet on the hydrophilic substrate is shorter than that on a hydrophobic substrate with heating.

Q6) Many of statements were to be expected or are obvious.

 (Answer) We tried to reflect the referee’s comment and revise our manuscript to improve the quality. We hope our manuscript could be published in Sensors.

Reviewer 2 Report

Dear authors,

Thank you for submitting your manuscript entitled 'Evaporation-rate Control of Water Droplets on Flexible Transparent Heater for Sensor Application' to MDPI Sensors. I think it's a wonderful work and of great interest to a broad readership. For a revised version, I would recommend to use vector graphics instead of pixel graphics. However, I recommend your manuscript for publication as is.

Author Response

Reviewer 2

Dear authors,

Thank you for submitting your manuscript entitled 'Evaporation-rate Control of Water Droplets on Flexible Transparent Heater for Sensor Application' to MDPI Sensors. I think it's a wonderful work and of great interest to a broad readership. For a revised version, I would recommend to use vector graphics instead of pixel graphics. However, I recommend your manuscript for publication as is.

 (Answer) We greatly appreciate the valuable recommendations and comments of the referee. As the referee’s comment, we used the vector graphics instead of pixel graphics.

Round 2

Reviewer 1 Report

Generally I accept the corrections and explanations of the authors as sufficient.